# Multi-Marker Approach in Patients with Acute Chest Pain in the Emergency Department

**DOI:** 10.3390/jpm14060564

**Published:** 2024-05-25

**Authors:** Andrea Piccioni, Silvia Baroni, Federica Manca, Francesca Sarlo, Gabriele Savioli, Marcello Candelli, Alessandra Bronzino, Marcello Covino, Antonio Gasbarrini, Francesco Franceschi

**Affiliations:** 1Department of Emergency, Anesthesiological and Reanimation Sciences, Fondazione Policlinico Universitario Agostino Gemelli-IRCCS, 00168 Roma, Italy; federica.manca@guest.policlinicogemelli.it (F.M.); marcello.candelli@policlinicogemelli.it (M.C.); alessandra.bronzino@policlinicogemelli.it (A.B.); marcello.covino@policlinicogemelli.it (M.C.); francesco.franceschi@policlinicogemelli.it (F.F.); 2Unit of Chemistry, Biochemistry and Clinical Molecular Biology, Department of Laboratory and Hematological Sciences, Fondazione Policlinico Universitario Agostino Gemelli-IRCCS, Università Cattolica del Sacro Cuore, 00168 Roma, Italy; silvia.baroni@policlinicogemelli.it (S.B.); francesca.sarlo@guest.policlinicogemelli.it (F.S.); 3Departement of Emergency, IRCCS Fondazione Policlinico San Matteo, 27100 Pavia, Italy; g.savioli@smatteo.pv.it; 4Medical and Surgical Science Department, Fondazione Policlinico Universitario Agostino Gemelli-IRCCS, Università Cattolica del Sacro Cuore, 00168 Roma, Italy; antonio.gasbarrini@policlinicogemelli.it

**Keywords:** multi-marker approach, suPAR, sST2, chest pain, hsTnI, acute coronary syndrome (ACS), emergency department (ED), early risk stratification

## Abstract

Background: Chest pain is a prevalent reason for emergency room referrals and presents diagnostic challenges. The physician must carefully differentiate between cardiac and noncardiac causes, including various vascular and extracardiovascular conditions. However, it is crucial not to overlook serious conditions such as acute coronary syndrome (ACS). Diagnosis of acute myocardial infarction (AMI) and early discharge management become difficult when traditional clinical criteria, ECG, and troponin values are insufficient. Recently, the focus has shifted to a “multi-marker” approach to improve diagnostic accuracy and prognosis in patients with chest pain. Methods: This observational, prospective, single-center study involved, with informed consent, 360 patients presenting to the emergency department with typical chest pain and included a control group of 120 healthy subjects. In addition to routine examinations, including tests for hsTnI (Siemens TNIH kit), according to the 0–1 h algorithm, biochemical markers sST2 (tumorigenicity suppression-2) and suPAR (soluble urokinase plasminogen activator receptor) were also evaluated for each patient. A 12-month follow-up was conducted to monitor outcomes and adverse events. Results: We identified two groups of patients: a positive one (112 patients) with high levels of hsTnI, sST2 > 24.19 ng/mL, and suPAR > 2.9 ng/mL, diagnosed with ACS; and a negative one (136 patients) with low levels of hsTnI, suPAR < 2.9 ng/mL, and sST2 < 24.19 ng/mL. During the 12-month follow-up, no adverse events were observed in the negative group. In the intermediate group, patients with hsTnI between 6 ng/L and the ischemic limit, sST2 > 29.1 ng/mL and suPAR > 2.9 ng/mL, showed the highest probability of adverse events during follow-up, while those with sST2 < 24.19 ng/mL and suPAR < 2.9 ng/mL had a better outcome with no adverse events at 12 months. Conclusion: Our data suggest that sST2 and suPAR, together with hsTnI, may be useful in the prognosis of cardiovascular patients with ACS, providing additional information on endothelial damage. These biomarkers could guide the clinical decision on further diagnostic investigations. In addition, suPAR and sST2 emerge as promising for event prediction in patients with chest pain. Their integration into the standard approach in PS could facilitate more efficient patient management, allowing safe release or timely admission based on individual risk.

## 1. Introduction

Chest pain is a frequent cause of admission to the ER [1,2,3,4] and can result from multiple origins, both cardiac and extracardiac [3,4,5]. It is crucial to quickly begin management of patients with life-threatening conditions such as acute coronary syndrome (ACS), aortic dissection, and pulmonary embolism, as well as other serious nonvascular conditions. Subsequently, targeted therapy must be given to those with less critical diseases. Although there are several life-threatening causes, chest pain often reflects more benign conditions [5].

Guidelines recommend an ECG within 10 min of arrival in the emergency department for patients with suspected acute myocardial infarction (AMI) [5,6,7]. High-sensitivity cardiac troponins are critical in identifying cardiac damage but do not specify the cause [7,8]. They may indeed be elevated in conditions such as acute heart failure, pericarditis, myocarditis, or arrhythmias [8]. However, a significant increase in cardiac troponins is an indicator of high risk for ACS [7,8,9].

The emergency physician must then stratify the patient’s risk for ACS or MACE in the short term. Prognostic tools such as HEART, GRACE, and TIMI scores are used to evaluate patients for whom ECG and troponin tests are not diagnostic [7,8,9,10,11,12,13,14]. According to these scores, patients with cardiac chest pain, non-elevated hsTnI, and a negative ECG for ACS, but with risk factors such as diabetes, hypertension, hypercholesterolemia, or smoking, are considered to be at intermediate risk and need further diagnostic tests. These may include stress testing, exercise imaging, CCTA, or ICA, which often prolong hospital stays [7].

Our goal is to propose a safe and efficient clinical pathway for intermediate-risk patients, allowing the clinician to safely discharge those who do not need further attention and focus on those at higher risk of ACS or MACE in the short term, using a multi-marker approach based on suPAR and sST2 values.

## 2. Biomarkers

Cardiac biomarkers are an essential component of the criteria used to establish the diagnosis of an acute myocardial infarction. Troponin is the biomarker of first choice for the detection of myocardial damage and has a class I indication for the diagnosis of myocardial infarction according to the most recent ESC guidelines [15].

The current limitation, but at the same time the strength, of high-sensitivity troponin assay methods is the possibility of detecting “biochemical positivity” that cannot always be ascribed to a context of myocardial ischemia but also of myocardial injury.

The high sensitivity of the troponin kits allows for the evaluation of a wide range of clinical conditions that are non-ischemic, both acute and chronic, cardiac and extracardiac, such as pericarditis, myocarditis, Takotsubo syndrome, tachyarrhythmias, heart failure, pulmonary embolism, stroke, and sepsis [8]. Thus, hs-troponins have a high specificity for “myocardial damage” with a high negative predictive value. In addition, their values should be interpreted as a quantitative marker of myocardial damage, as there is a direct proportionality between the troponin level and the extent of injury.

SST2 has recently emerged as a promising biomarker in the field of acute cardiovascular disease. ST2 is a protein that belongs to the interleukin-1 receptor family (IL-1 RL-1) and exists in two isoforms, one transmembrane (ST2L) and one soluble (sST2). The natural ligand of ST2 is IL-33, a member of the IL-1 family, which plays an anti-inflammatory, anti-hypertrophic, and anti-fibrotic role in the myocardium [16] and is produced by different cell types, such as endothelial cells.

sST2, the soluble variant of ST2, on the other hand, is over-expressed under specific pathological conditions of myocardial stress or damage and is associated with inflammation and immune response; it is a decoy receptor that reduces the cardioprotective effect of the IL-33/ST2L pathway by binding free IL-33, which will no longer be available to the ST2L receptor [17].

In fact, sST2 is released by myocardial cells in response to myocardial infarction [18].

Although sST2 does not add elements to the initial diagnosis of AMI, its prognostic role has become conspicuous: early levels of sST2 during AMI seem to reflect the extent of myocardial necrosis [19]. Indeed, although sST2 cannot replace natriuretic peptides (NPs) in the diagnosis of AHF, its baseline increase has been shown to be superior to NT-pro BNP as a prognostic marker of mortality at 1 year and 4 years [19].

The role of ST2 in the pathophysiology of CAD and the clinical value of this biomarker in acute ST-segment elevation myocardial infarction (STEMI) have broadly expanded. Several studies suggest an important prognostic value of sST2 [20,21,22], both in chronic heart failure [23], in which it predicts the outcome of patients in addition to the N-terminal pro-B type natriuretic peptide (NT-proBNP) and the highly sensitive troponin T [24], and in acute heart failure [25], where it is useful in monitoring and guiding therapeutic decisions in patients with acute decompensated heart failure and acute myocardial infarction [26]. As a biomarker related to inflammation and fibrosis, sST2 has important clinical value in CAD, which may guide prognosis prediction and treatment [25,27,28].

In clinical practice, sST2 can be used in the prognostic stratification of patients and for the identification of patients at high risk of mortality and re-hospitalization in patients diagnosed with heart failure [29], providing unique prognostic information as well as complementary to that provided by the natriuretic peptides BNP and NT-proBNP, from which it is independent [30].

Although sST2 does not add elements to the initial diagnosis of AMI, its prognostic role has become conspicuous. Early levels of sST2 in AMI appear to reflect the extent of myocardial necrosis [19]. Starting from these data, sST2 could be recognized as a marker of early and late post-infarction remodeling [26,27,28,29,30,31].

SuPAR is the soluble form of uPAR, the membrane receptor of urine-type plasminogen activator or urokinase (uPA), and is released from the plasma membrane upon cleavage from the GPI anchor in response to inflammatory stimuli, regardless of the underlying cause [32].

Serum levels of suPAR are thus closely related to immune and inflammatory activation, and in the context of cardiovascular disease, suPAR has emerged as a very promising biomarker as a prognostic indicator for early prediction of events in chest pain emergency patients [33].

In fact, although its non-cardiac-specific nature limits its diagnostic value for heart disease, when used in conjunction with imaging studies and clinical rating scales in a multi-marker approach, it has been shown to ameliorate the clinical capacity to identify patients at risk for adverse cardiovascular events, morbidity, and mortality [34]. In the setting of an ACS, suPAR is associated with long-term all-cause death, heart failure, and MACE and provides incremental prognostic value beyond traditional risk factors [35].

## 3. Endpoint

The study sought to investigate a multi-marker approach in patients presenting to the emergency department (ED) with chest pain suggestive of acute coronary syndrome (ACS), using sST2 and suPAR, in addition to hsTnI, in the diagnostic workup and early risk stratification, to assess whether this approach improves diagnostic or prognostic accuracy, especially in those patients without a clear diagnosis. In addition, the effectiveness and validity of the multi-marker approach were evaluated through a follow-up of our patients at 12 months after ED admission.

## 4. Methods

### 4.1. Study Design

We performed a monocentric and prospective study evaluating a multi-marker approach for patients presenting cardiac pain in the Emergency Department. This study was conducted and developed in close cooperation with the High Automation Corelab and the Emergency Medicine Department of the A. Gemelli University Polyclinic Foundation, IRCCS of Rome. We prospectively and consecutively enrolled a total of 360 patients who presented to the ED of our polyclinic with chest pain or other typical symptoms. The inclusion criteria for the patient group were: (1) age ≥ 18 years; (2) chest pain or suggestive symptoms suspecting ACS; and (3) ability to give informed consent. Patients only presenting with dyspnoea or palpitations were not included in our study. All enrolled patients underwent standard diagnostic and therapeutic procedures according to the updated guidelines and good clinical practices. There were no alterations to the approved ACS protocol by our Polyclinic Foundation.

High-sensitivity troponins were determined following the European Society of Cardiology’s (ESC) 0/1-h protocol.

Baseline demographic and clinical data (age, gender, cardiovascular risk factors, blood pressure, heart rate, electrocardiogram, respiratory rate, arterial oxygen saturation) as well as information on medications were collected and recorded on standardized data collection forms (see Table 1).

Plus, multi-markers’ measurements (sST2 and suPAR) were performed for each patient enrolled, using the same blood sample provided by the clinical care pathway for chest pain approved in our hospital. Blood samples were immediately stored in the High Automation CoreLab of our hospital; plasma and serum were aliquoted in Eppendorf tubes and frozen at −80 °C until sST2 and suPAR analysis. The levels of all biomarkers were also examined in a control group of 120 healthy subjects with homogeneous characteristics regarding sex and age [35,36,37].

In our study, we did not utilize the GRACE or TIMI score, as we aimed to enhance prognostic value with actual measurements rather than clinical management tools.

### 4.2. Biomarker Measurements

Biomarker measurements were performed in the High Automation CoreLab of the A. Gemelli University Polyclinic Foundation, IRCCS Rome”, by an Atellica Solution analyzer (Siemens Healthineers, Malvern, PA, USA).

-hs-troponin was measured by the TnIH kit (Siemens Healthcare Diagnostics, USA) using the CLIA method. The limit of detection (LOD) is 2.5 ng/L; the 99th percentile cut-off is 57 ng/L for males and 37 ng/L for females, with 10% CV at 6 ng/L. The TNIH assay-specific cut-off level (6 ng/L) within the 0 h/1 h protocol was derived from pre-defined criteria for sensitivity and specificity for ASC, as reported in ESC guidelines 2023 [15].-sST2 was measured by the Sequent-IA kit, Critical Diagnostics USA, using the turbidimetric method applied to Atellica CH Siemens [38]. The LOD is 8 ng/mL, the measuring range is from 8 to 360 ng/mL, and the cut-off value for heart failure risk is 35 ng/mL.-suPAR was measured by the turbidimetric method with the SUPARNOSTIC kit (Virogates DK) applied to Atellica CH Siemens. The LOD is 1.7 ng/mL, and the range is from 1.7 to 26.5 ng/mL; the cut-off is 3.0 ng/mL.

### 4.3. Follow-Up

Follow-up data were retrieved from digital and written patients’ records, including discharge letters, revascularization reports, and any other relevant documentation. The 12-month clinical follow-up data were obtained from all patients included in the study by phone interview to assess clinical outcome (symptoms such as chest pain), hospital re-admission (myocardial infarction, revascularization), coronary angiography, and death.

### 4.4. Ethics Statements

The study protocol did not alter the diagnostic workup or therapeutic management of enrolled patients in any way. The study was approved by the A. Gemelli University Polyclinic Foundation, IRCCS, Ethics Review Board (protocol n° 4896/22). All patients provided written informed consent for their participation in the study.

### 4.5. Statistical Analysis

Statistical analysis was conducted using MedCalc^®^Statistical version 19.5.6 (MedCalc Software Ltd., Ostend, Belgium; https://www.medcalc.org (accessed on 16 May 2024); 2020) (version 15.0) software. Continuous variables were expressed as mean values ± standard deviation or as median (range), and categorical variables were expressed as frequencies. The data distribution was assessed by the Kolmogorov–Smirnov test or the Shapiro–Wilk test in order to verify the population distribution. The most appropriate statistical parametric and non-parametric tests (the Student’s *t*-test or Mann–Whitney and chi-squared or Fisher) were used based on the data distribution. The Youden Index was used to choose the optimal cut-off points for biomarkers’ diagnostic and prognostic effectiveness. The correlation analysis between variables was carried out using the Spearman coefficient. We calculated odds ratios (ORs) with their respective 95% confidence intervals (95% CI). A *p* value < 0.05 was considered statistically significant.

## 5. Results

Table 2 reports baseline values of hsTnI, sST2, and suPAR in all patients (N = 360).

Then, we identified our study population in 2 groups:-The “healthy control” group, composed of 120 subjects with hsTnI < 2.5 ng/L, in which levels of sST2 and suPAR were ≤ 2 4.19 ng/mL and ≤ 2.9 ng/mL, respectively;-The “true positives” group, composed of 112 patients with hs-cTn concentration at presentation at least moderately elevated above the ischemic cut-off level (57 ng/L for males and 47 ng/L for females) or with a significant rise within the first hour (1 hΔ) of hsTnI levels.

The ACS diagnosis was confirmed, and all patients in this group had sST2 levels above 24.19 ng/mL and suPAR levels above 2.9 ng/mL.

We identified that sST2 and suPAR levels were significantly different in true positives versus the healthy group (*p* < 0.001); ST2 levels above 24.19 ng/mL had 100% specificity and 100% sensitivity for ACS, while suPAR levels above 2.9 ng/mL had 100% sensitivity and 85.7% specificity (see Figure 1 and Figure 2). These cut-offs were calculated with Youden’s index.

Figure 1 and Figure 2 display the ROC for sST2 (Figure 1) and suPAR (Figure 2) in the *true positives* vs. *healthy group* (cut-off of 24.19 ng/mL for sST2 and 2.9 ng/mL for suPAR).

Then, we divided the patients according to hsTnI levels into a “*negative*” group composed of 136 patients with hsTnI below 6 ng/L and an “*intermediate*” group composed of 112 patients with troponin levels > 6 ng/L but below the ACS diagnostic cut-off for hsTnI (< 47 /57 ng/L F/M).

According to the suPAR normal cut-off, we observed that in the “*negative*” group, patients with suPAR ≤ 2.9 ng/mL (n 59) showed all sST2 levels ≤ 24.19 ng/mL, while patients with suPAR > 2.9 ng/mL (n 77) showed sST2 > 29.1 ng/mL in 65% of the cases. In the “*intermediate*” group, patients with suPAR ≤ 2.9 ng/mL (n 52) had sST2 ≤ 24.19 ng/mL, while all patients with suPAR > 2.9 ng/mL had sST2 > 29.1 ng/mL.

After 12 months from ED admission, it was possible to obtain information about the patients’ follow-up through a telephone interview; we considered any event reported that we defined as MACE to be a negative outcome, regardless of the score (presence of single or multiple events).

sST2 and suPAR’s cut-offs, previously calculated with ROC curves, were related to the presence or absence of adverse events of cardiac origin at the one-year follow-up through the chi-square “χ^2^” hypothesis test and the relative risk.

We observed that in the *negative* and *intermediate groups,* none with suPAR ≤ 2.9 ng/mL and sST2 < 24.19 ng/mL reported adverse events in the follow-up (specificity 100%); these patients seem like healthy subjects.

While in the *negative* group, patients with suPAR > 2.9 ng/mL and sST2 > 29.1 ng/mL reported events in 70% of the cases.

In the *intermediate group*, almost all of the patients (84%) with sST2 > 29.1 ng/mL had adverse events at 1 year, while none showed events if sST2 was lower than 24.19 ng/mL.

The ROC curve AUC of sST2 for the excluding of events was 0.974 at the optimal threshold of 24.19 ng/mL, with a sensitivity of 100% and a specificity of 87.5% (Figure 3). The ROC curve AUC of sST2 for the prediction of events was 0.852 at the optimal threshold of 29.1 ng/mL, with a sensitivity of 85.7% and a specificity of 81.8% (Figure 4).

In the *intermediate group*, according to the suPAR 2.9 ng/mL cut-off, the sensitivity was 86% and the specificity was 95.7% for identifying patients at risk or not for adverse events at 1 year. In fact, 80% of patients with values higher than 2.9 ng/mL showed adverse events at 1 year, while only 2% of patients with values lower than 2.9 ng/mL had adverse events in the follow-up.

sST2 was highly sensitive and specific (100% sensitivity, 90.9% specificity, positive predictive value 83.8%, negative predictive value 100%) for adverse events during follow-up; values above the cut-off of 29.1 ng/mL were therefore indicative of myocardial distress and associated with patients with a positive outcome.

suPAR was specific and highly sensitive (98.02% sensitivity, 74.5% specificity, positive predictive value 80%, negative predictive value 97.6%) for adverse events during follow-up with values above the cut-off of 2.9 ng/mL.

We also verified the significance of the association of sST2 and suPAR levels with outcome using the “χ^2^” chi-square hypothesis test, which showed high statistical significance (*p* value < 0.001). We calculated the relative risk of an adverse event at one year in the case of sST2 higher than the cut-off of 29.1 ng/mL (χ^2^ = 70.66, RR = 51.11, 95% CI 7.28–358.87, *p*-value < 0.001) and in the case of suPAR higher than the cut-off of 2.9 ng/mL (χ^2^ = 63.28, RR = 3.86, 95% CI 2.45–6.07, *p*-value < 0.001).

## 6. Discussion

Patients presenting to the ED with acute chest pain are a challenge for emergency physicians and acute medicine departments, as a wide spectrum of diagnoses may cause the pain, ranging from acute myocardial infarction (AMI) and pulmonary embolism to harmless muscular tension belonging to the group of chest wall syndromes, as well as gastrointestinal causes such as gastroesophageal reflux disease. Noncardiac causes are very common, but it is important not to overlook serious conditions.

The decision to admit or discharge a patient can be difficult for the physician, especially in an acute setting such as the ED, which requires making this decision within hours. Currently, the evaluation of patients consists of a clinical investigation concerning medical history, a physical examination, followed by a 12-channel electrocardiogram (ECG), and further focused diagnostics, including cardiac biomarkers.

Despite the development of improved cardiac biomarkers and the validation of clinical scoring systems, medical decisions about hospital admission or discharge do not always come promptly. The introduction of highly sensitive cardiac troponin (hs-cTn) has significantly improved the diagnostic accuracy of acute coronary syndrome, both in the rule-in phase (with anticipation of the diagnosis) and in the rule-out phase, with the possibility of excluding acute injury with a 1-hour blood sample. On the other hand, hsTnI can highlight any kind of myocardiocyte suffering, not only that due to an ischemic cause, thus adding doubts to the decision-making process in which the physician must choose between hospital admission, 24-h observation, or a safe discharge. Recently, the standardization and validation of new diagnostic methods have increased clinicians’ confidence in the use of circulating biomarkers to improve diagnostic accuracy, assess individual risk of developing cardiovascular disease, and monitor associated adverse events.

sST2 has been studied in patients with heart failure and has recently been introduced as an additional prognostic marker for natriuretic factors in heart failure, as it increases when myocardial tissue remodels in fibrosis. Plus, because of sST2’s increase in different pathophysiological cardiac pathways, other studies have highlighted that sST2 has a complementary role in the prognostic stratification of cardiac ischemic patients. While hsTnI’s increase represents miocardiocyte suffering, the sST2 plasma level may reflect endothelial damage.

Recently, many studies have taken into consideration suPAR as a cardiovascular marker, as it is related to immune activation, inflammation, and endothelial damage. suPAR has been proposed as a biomarker for risk stratification and for monitoring the therapeutic response in patients with heart disease.

Our research is based on a multi-marker approach for patients who are admitted to the ED with acute chest pain. We enrolled 360 consecutive patients in whom ACS was suspected. We measured hsTnI 0–1 h and the other biomarkers with the same blood sample. The aim of our research was to evaluate whether the combination of different biomarkers could help the clinician find an accurate diagnosis, explain the origin of the patients’ symptoms, and improve risk stratification and prognosis. From our statistical analysis, we noticed that the distribution of values was very heterogeneous, considering the multiple clinical features that can be responsible for acute chest pain.

HsTnI was the only biomarker able to vary between time 0 and time 1, as its immediate release was an expression of an acute event; the other biomarkers did not show significant variation between the two measurements.

Our findings confirmed that troponin has a diagnostic role and highlighted that suPAR and sST2 could help the clinician better understand and personalize each patient’s cardiovascular risk.

According to troponin levels, we divided the patients into different groups.

To evaluate the reliability of a multi-marker approach in these patients, we divided the *negative* subjects on the suPAR level: suPAR ≤ 2.9 ng/mL and sST2 ≤ 24.19 ng/mL showed high accuracy for excluding ACS, while suPAR > 2.9 ng/mL and sST2 > 29.1 ng/mL could discriminate patients with ACS.

The group of patients with “*intermediate*” troponin is the group that usually creates the biggest problems for the clinician. Patients with sST2 > 29.1 ng/mL or suPAR > 2.9 ng/mL presented a high risk of ACS. On the other hand, patients with sST2 ≤ 24.19% and suPAR ≤ 2.9 ng/mL were found to be healthy, non-ACS patients.

Among patients with *negative* or *intermediate* troponin, sST2 and suPAR appear to be able to separate those who could be discharged safely from the ED from those who may need more attention.

For follow-up, we relied mainly on objective data such as re-hospitalization, treatment changes, and new visits to the PS, rather than exclusively on the limited value of the 12-month telephone analysis. This is because patients’ re-presentation of symptoms often lacks accuracy and timeliness. Follow-up showed that ST2 > 29.1 ng/mL was highly specific for adverse events and thus associated with patients with adverse outcomes, and suPAR > 2.9 ng/mL was specific and highly sensitive for adverse events.

For the first time in the evaluation of patients with acute chest pain, our study suggests the use of a multi-marker approach with sST2 and suPAR in association with troponin. Troponin, sST2, and suPAR explore totally distinct pathways that can each independently contribute to the genesis of cardiovascular pathology, despite being able to coexist without a causal relationship in some cases. Furthermore, sST2 and suPAR must also be considered indicators of endothelial health, as they are strongly influenced by conditions such as inflammatory states, autoimmune diseases, neoplasms, and non-ischemic heart conditions. SuPAR and sST2 are promising biomarkers for early prediction of events in chest pain emergency patients. Our data are preliminary and require an expansion of patients to be enrolled. However, since clinicians increasingly ask the laboratories for support in guiding their diagnostic choices using an evidence-based, biomarker-based approach, our findings appear interesting and deserve to be carefully considered and evaluated so as to manage patients more efficiently in a difficult setting such as the emergency department.

## 7. Limitations

There are some limitations to our study that are worth taking into consideration. First, the study is limited by a small sample size, which may influence our findings. However, our research should be viewed as a starting point for a multi-center study with a larger sample size. Plus, multi-center studies are needed to determine the potential role of this multi-marker approach in the diagnostic work-up as well as in patient stratification for rule-in and rule-out. Second, we lacked serial measurement of biomarkers in our study; the plasma levels of sST2 and suPAR were only detected at the time of admission to the ED. Temporal variation in biomarker measurements from admission to discharge in the ED could be an important finding. However, despite these limitations, it is important to underline our great findings regarding this multi-marker approach in patients with chest pain. The follow-up is still in progress.

## 8. Conclusions

Our data suggest that hsTnI remains the biomarker of choice for acute cardiological evaluations, considering its peculiar cardio-specificity and its rapid increase in acute ischemic settings; on the other hand, it is known that numerous different pathophysiological elements can influence each other and associate, determining multiple clinical pictures in patients. SST2 may play a complementary role to troponin in the prognostic stratification of ischemic patients. SuPAR, as a marker of endothelial damage and involvement in various pathophysiological pathways, can guide clinicians in determining the need for further diagnostic investigations. Our data suggest that a future integration of these biomarkers into the routine approach to the patient with acute chest pain in the ED might allow better patient stratification and proper patient management, helping the clinician to make an early safe discharge or a timely admission for those who deserve in-depth diagnostic and therapeutic investigations.

## Figures and Tables

**Figure 1 jpm-14-00564-f001:**
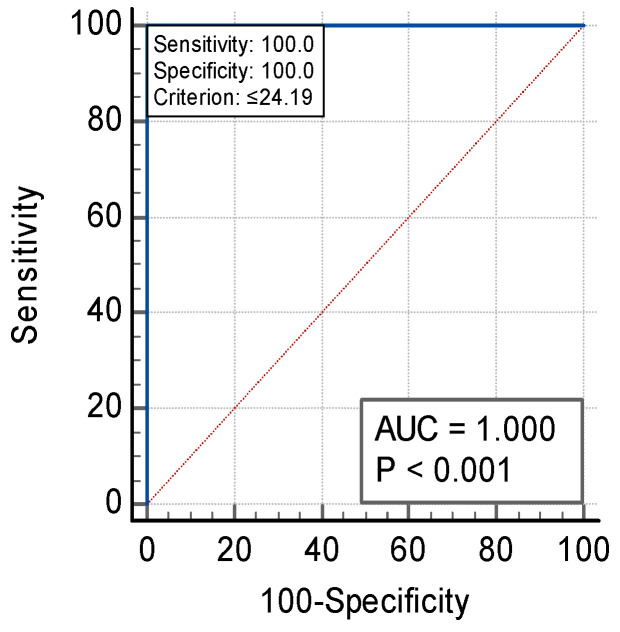
ROC for sST2.

**Figure 2 jpm-14-00564-f002:**
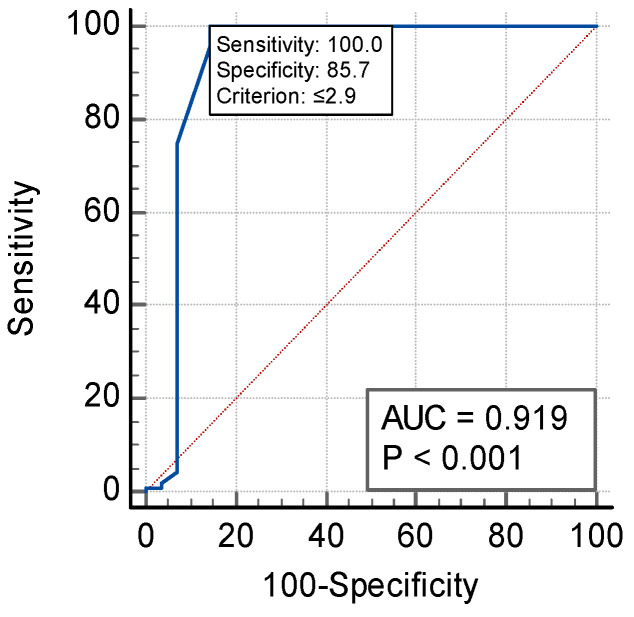
ROC for suPAR.

**Figure 3 jpm-14-00564-f003:**
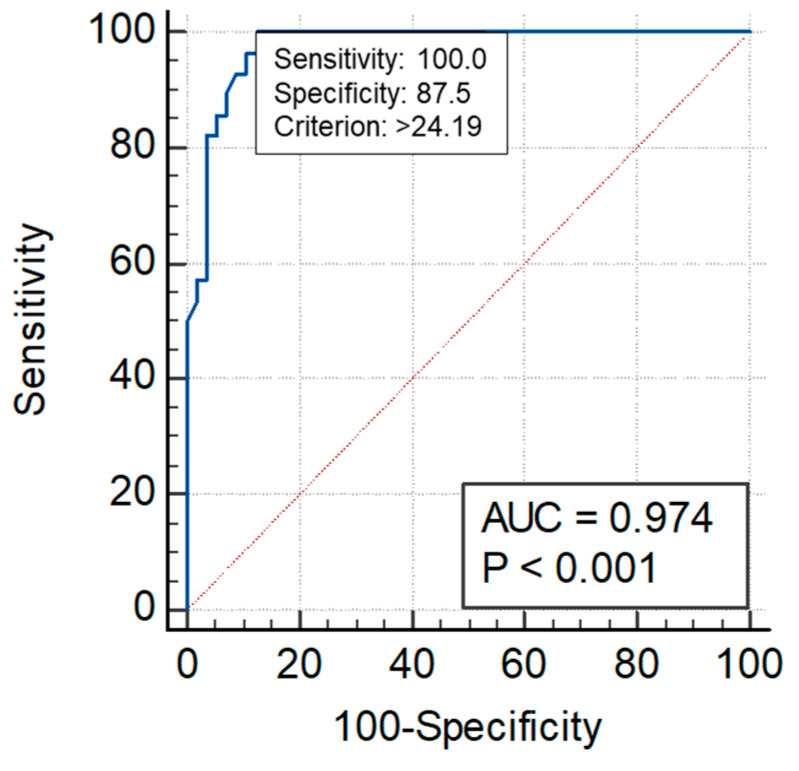
The *intermediate group* sST2 cut-off for the excluding of events (24.19 ng/mL).

**Figure 4 jpm-14-00564-f004:**
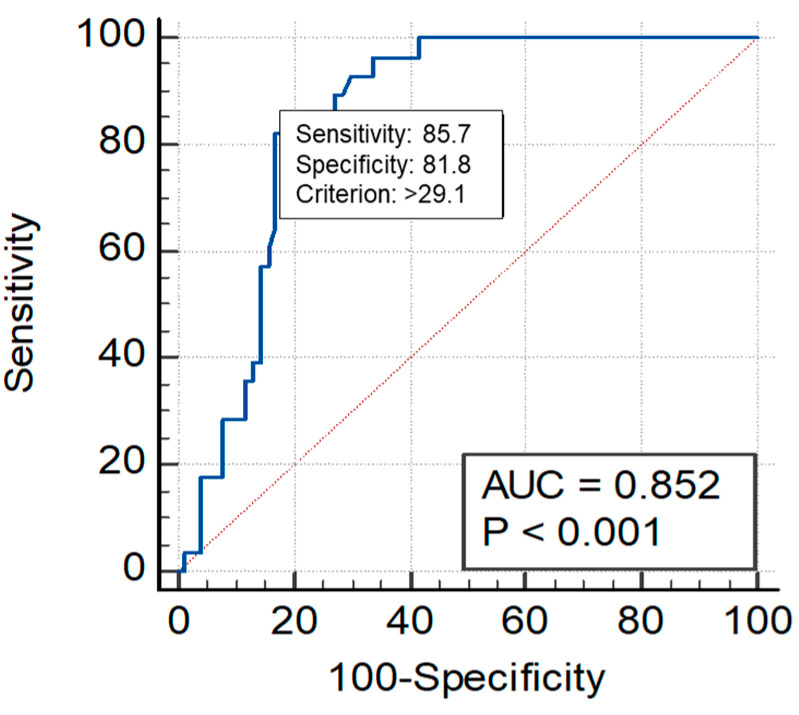
The *intermediate group* with a high risk of adverse events for the sST2 cut-off of 29.1 ng/mL.

**Table 1 jpm-14-00564-t001:** Baseline characteristics of the studied population (N = 360 patients).

Characteristics	%
N pts	360
Sex (M:F)	57.6% 42.4%
Age (aa)	56.7 aa (33–86) M 215 (33–86) F 145 (36–77)
Risk factors	
Hypertension	52.8% M 58.3% F 45.2%
Diabetes	16.8% M 20.8% F 11.3%
Dyslipidemia	36.4% M 36.1% F 36.8%
Familiarity with cardiovascular disease	22.4% M 22.9% F 21.7%
Smoke	16.4% M 22.2% F 8%
Vasculopathy	10.4% M 14.6% F 4.7%

**Table 2 jpm-14-00564-t002:** Baseline biomarker levels in the patients enrolled (N = 360).

	Min	Max	Mean	Median	SD	25–75 P
Age (y)	23.0	90.0	64.47	66.0	15.3	55.0 to 76.0
hsTnI ng/L	2.5	87,239.0	850.64	6.0	7353.5	3.0 to 21.0
sST2 ng/mL	12.0	501.6	34.48	24.8	48.6	19.1 to 37.5
su-PAR ng/mL	1.70	26.5	4.07	3.5	2.8	2.7 to 4.3

## Data Availability

The original contributions presented in the study are included in the article, further inquiries can be directed to the corresponding author.

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
