# Peer review of "Multi-Marker Approach in Patients with Acute Chest Pain in the Emergency Department"

_jpm, 2024, doi:10.3390/jpm14060564_

Round 1

Reviewer 1 Report

Comments and Suggestions for Authors

I read an interesting study related to the role of biomarkers regarding the prognosis of patients with chest pain.

Your research is focused on the prognostic role of the 2 new markers (sST2 and suPAR) in patients with ACS. However, I would like you to discuss in more detail the results obtained by you regarding the negative predictive value for the initial diagnosis of ACS and the data from the literature regarding the diagnostic utility of the 2 new biomarkers.

I think it would be useful to add (in the "Discussions" part) the limited value of the telephone evaluation at 12 months regarding the cardiac events of the followed patients.

Also pay attention to some small mistakes: page 8, line 262, probably "negative outcome", instead of "positive outcome", as well as the repetition in the Bibliography of the same source at 26 and 33 (Maisel A.S., Filippatos G.S. Algorithms in Heart Failure. Jaypee, The Health Sciences Publisher; New 458 Delhi, India: 2016).

With these additions, I think it's good for publishing.

Author Response

Dearly beloved, we thank you for your suggestions and corrections. As requested, below are the point-by-point changes in the revision. 

We remain available for further clarification or necessary changes.

Reviewer 1

  1. Your research is focused on the prognostic role of the 2 new markers (sST2 and suPAR) in patients with ACS. However, I would like you to discuss in more detail the results obtained by you regarding the negative predictive value for the initial diagnosis of ACS and the data from the literature regarding the diagnostic utility of the 2 new biomarkers.

Yes, it has been done. The focus of the research was on studying the prognostic significance of biomarkers rather than their use for diagnostic purposes, as specified repeatedly in the manuscript.

  1. I think it would be useful to add (in the "Discussions" part) the limited value of the telephone evaluation at 12 months regarding the cardiac events of the followed patients.

It has been added as recommended.

  1. Also pay attention to some small mistakes: page 8, line 262, probably "negative outcome", instead of "positive outcome", as well as the repetition in the Bibliography of the same source at 26 and 33 (Maisel A.S., Filippatos G.S. Algorithms in Heart Failure. Jaypee, The Health Sciences Publisher; New 458 Delhi, India: 2016).

The changes were made.

------------------------------------------------------

Reviewer 2 Report

Comments and Suggestions for Authors

I congratulate the authors for their work on this manuscript, as it is of great value for the clinical practice. 

Here are my comments: 

- the abstract is too long and needs rephrasing. It should encorporate only the key ideas of the manuscript and not the entire design, as it should encourage the reader to read the entire manuscript. 

- the introduction is also too long, maybe some phrases could be moved to the discussion part.

- there are parts of the article that have different fonts, spacing and size. Please revise that.

- some phrases lack the final "." - 158,204, 208,211, 227, 307 etc. - please check

Comments on the Quality of English Language

Minor english revision is needed throughout the manuscript

Author Response

Dearly beloved, we thank you for your suggestions and corrections. As requested, below are the point-by-point changes in the revision. 

We remain available for further clarification or necessary changes.

Reviewer 2

  1. the abstract is too long and needs rephrasing. It should encorporate only the key ideas of the manuscript and not the entire design, as it should encourage the reader to read the entire manuscript.

The abstract has been reworded.

  1. the introduction is also too long, maybe some phrases could be moved to the discussion part.

The introduction has been revised and edited.

  1. there are parts of the article that have different fonts, spacing and size. Please revise that.

The article has been fixed.

  1. some phrases lack the final "." - 158,204, 208,211, 227, 307 etc. - please check

The errors have been corrected.

  1. Minor english revision is needed throughout the manuscript.

The changes have been made.